# Reductions in prefrontal activation predict off-topic utterances during speech production

Paul Hoffman [1]

The ability to speak coherently is essential for effective communication but declines with age: older people more frequently produce tangential, off-topic speech. Little is known, however, about the neural systems that support coherence in speech production. Here, fMRI was used to investigate extended speech production in healthy older adults. Computational linguistic analyses were used to quantify the coherence of utterances produced in the scanner, allowing identification of the neural correlates of coherence for the first time. Highly coherent speech production was associated with increased activity in bilateral inferior prefrontal cortex (BA45), an area implicated in selection of task-relevant knowledge from semantic memory, and in bilateral rostrolateral prefrontal cortex (BA10), implicated more generally in planning of complex goal-directed behaviours. These findings demonstrate that neural activity during spontaneous speech production can be predicted from formal analysis of speech content, and that multiple prefrontal systems contribute to coherence in speech.

[1] School of Philosophy, Psychology and Language Sciences, University of Edinburgh, Edinburgh EH8 9JZ, UK. Correspondence and requests for materials should be addressed to P.H. (email: p.hoffman@ed.ac.uk)

Although it often appears effortless, production of conversational speech involves complex cognitive processes. In order to speak coherently on a topic, individuals must constantly regulate their speech output to ensure that they produce statements that are informative and relevant to the topic under discussion. Numerous studies have found that this ability declines in later life, with changes evident in people aged 60 and above[1,2]. Older adults are more likely to produce tangential, off-topic utterances during conversation[3,4] and to provide irrelevant information when telling a story[2] or describing an object[5]. Although these effects have been observed in a range of tasks, age-related coherence declines are most pronounced when individuals provide information from their own memory or personal experience, and less severe when they describe visually presented stimuli, such as pictures or comic strips[1,6,7].

Loss of coherence can reduce the effectiveness of communication and the quality of older people's verbal interactions[8]. Less coherent speech is associated with higher levels of stress and less satisfaction in social interactions[4,8,9]. Researchers have often made a distinction between local coherence, the degree to which adjoining utterances related meaningfully to one another, and global coherence, the degree to which each utterance relates to the topic under discussion[3,10]. Speech that is tangential or off-topic is therefore said to be low in global coherence, while speech that shifts abruptly between subjects is low in local coherence (although these measures are typically correlated). As most studies have reported that global coherence declines more severely than local coherence in later life[2,3,7], this aspect of coherence was the focus of the present work.

Despite its importance for effective communication, there is limited understanding of the cognitive and neural mechanisms involved in maintaining coherence during speech. Explanations for the age-related decline in coherence have focused on changes to domain-general cognitive control processes, rather than aspects of the language system per se[1]. A number of studies have reported that more coherent individuals perform better on tests of cognitive or executive control, such as the Stroop task[1,4,7,9]. One view is that declines in coherence result from a reduced ability to inhibit irrelevant information, which means that older people are less able to prevent tangential or off-topic ideas from intruding into their discourse[4,11]. Supporting this idea, a recent behavioural study demonstrated that the ability to select task-relevant aspects of semantic knowledge is a strong determinant of coherence[12].

Less is known about the brain regions involved in promoting on-topic, coherent utterances during speech. This is in part due to a dearth of functional neuroimaging studies probing speech production beyond the single-sentence level. Collection of fMRI data during extended speech production has sometimes been considered problematic due to signal contamination caused by excessive head and jaw movements[13]. A small number of neuroimaging studies have overcome these technical challenges, however. These indicate that, in addition to engaging areas involved in motor planning and production, extended speech production activates an left-lateralised network including ventral temporal and inferior parietal regions involved in the representation of semantic knowledge and prefrontal regions associated with planning and cognitive control[14–18]. The majority of participants in these studies were healthy young adults, however; no studies to date have used fMRI to investigate extended speech production in older people specifically.

Perhaps more importantly, no studies have investigated how brain activity during coherent speech differs from that associated with off-topic, incoherent speech. Identifying the neural predictors of coherence is critical if we are to understand the root causes of age-related declines in the ability to produce coherent speech. To address this issue, in the present study a group of older people were scanned with fMRI while they spoke about a series of topics for 50 s at a time. To ameliorate the effect of speech-related head movements, principal components analysis was used to partition genuine haemodynamic signal changes from movement-related artefact[19,20] for similar approaches, see ref. [21]. Speech produced in the scanner was recorded, transcribed and subjected to computational linguistic analyses which quantified its coherence at each time-point during the scanning. This allowed investigation of (a) regions activated by older people during extended speech production, relative to a rote speech baseline and (b) regions whose activity varied as a function of the coherence of the speech produced. Thus the analysis identified brain regions that support coherence for the first time. The main steps in the study are illustrated in Fig. 1.

The study predictions stemmed from the idea that to speak coherently, people must regulate their access to semantic knowledge so that they select the most relevant information to drive speech output. According to long-standing models of language comprehension, when people *comprehend* speech or text they generate a mental model of its content, often termed a situation model[22–24]. This situation model is informed by the individual's prior semantic knowledge about the topic under discussion. It is likely that a similar model-building process guides speech production[10,25,26]. In order to remain coherent, speakers therefore need to ensure that currently relevant semantic knowledge contributes to the situation model guiding their production, while inhibiting irrelevant aspects of knowledge, which may lead to tangential or off-topic speech[4,11,27]. The key prediction was therefore that coherence in speech would be correlated with activation in brain regions that regulate access to semantic knowledge, chiefly the left inferior frontal gyrus (IFG).

Left IFG has been identified as the key node in a network of regions that control the retrieval and selection of semantic information[28,29]. This area is thought to exert top-down regulation over the activation of semantic knowledge, based on current contextual and task demands[30,31]. rTMS applied to posterior left IFG (BA44) has been shown to impair coherence in healthy young adults, suggesting that this region also contributes to the regulation of speech output[32]. However, different functions have been proposed for discrete regions within IFG[33,34]. Pars orbitalis (BA47) is thought to support controlled retrieval of weaker or less salient semantic associations from memory, while pars triangularis (BA45) is involved in selecting the most currently relevant aspects of retrieved knowledge and inhibiting activation of irrelevant knowledge (semantic selection). In line with the theoretical perspective outlined above, a recent behavioural study indicated that semantic selection ability is a strong predictor of coherence[12]. People who are skilled at selecting the most relevant aspects of their knowledge tend to be highly coherent when speaking because they are able to avoid producing tangential, off-topic information. For this reason, I predicted that activation of left BA45 in particular would be associated with production of highly coherent speech.

In addition, the study tested whether left and right IFG showed similar relationships with coherence. In young adults, IFG activation during verbal semantic processing is strongly left-lateralised, while bilateral IFG activation is observed more frequently in older people[35]. There is ongoing debate as to whether bilateral recruitment of IFG in later life is an adaptive strategy that boosts cognitive performance or whether it represents unhelpful dedifferentiation in neural activity[36–39]. In light of this debate, the present study tested whether IFG involvement in coherent speech was bilateral in older people and whether more coherent older speakers displayed greater recruitment of right IFG.

The study finds that highly coherent speech is associated with increased activity in inferior prefrontal cortex (BA45) bilaterally,

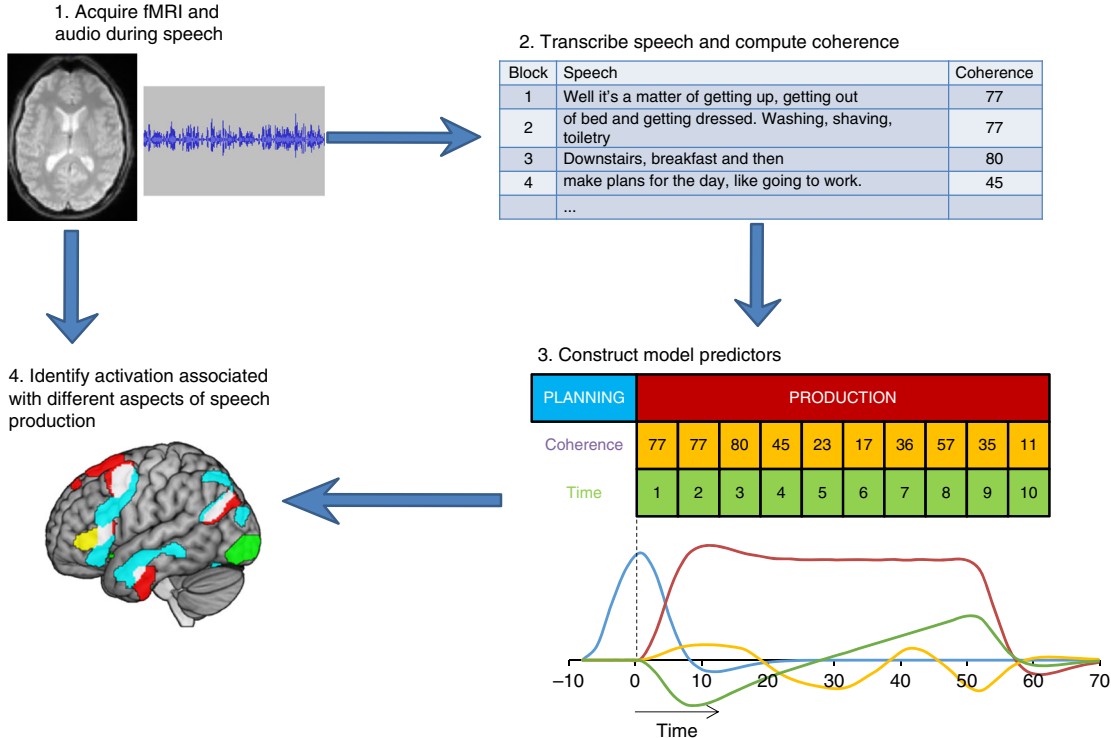

**Fig. 1** Stages in data analysis. 1. Participants were recording while speaking in the scanner. 2. Speech was divided into blocks of 5 s, transcribed and analysed using computational linguistic techniques to provide an estimate of coherence for each 5 s block. 3. A general linear model was constructed with regressors tracking speech planning periods, speech production periods, activation correlated with coherence and activation changing over time. 4. The model was fitted to voxel time-series in whole-brain and ROI analyses

as well as in bilateral rostrolateral prefrontal cortex (RLPFC; BA10), an area implicated in planning of complex goal-directed behaviours. Coherence-related activity in RLPFC precedes that in left BA45. These findings indicate that multiple prefrontal systems make distinct contributions to coherence in speech. At a more general level, they demonstrate that neural activity during spontaneous speech production can be predicted from formal analysis of speech content.

## Results

**Characteristics of speech production.** Participants produced an average of 104 words in response to each prompt (range: 50–153). Global coherence was computed for each response, using a computational linguistic approach (see Methods for details). The mean coherence value for responses was 47.6 (range: 20.9–81.4), which is similar to levels observed when speech is produced out of the scanner[12]. As described in Methods, the coherence measure quantifies the strength of the semantic relationship between the speech produced and the typical responses made to the same prompt. It uses a cosine similarity metric, which varies between 0 and 1 and is multiplied by 100 here for ease of presentation. 0 therefore indicates speech that has no semantic relationship with the topic being probed and 100 indicates speech that is identical to it.

There was considerable variation in coherence across individuals: the least coherent individual had a mean coherence score of 41.5 and the most coherent 55.0. Importantly, the coherence measure showed high test-retest reliability across individuals. Fourteen of the participants had previously completed a similar speech elicitation task out of the scanner as part of an earlier study[12]. Their in-scanner and out-of-scanner coherence scores were highly correlated ($r = 0.88$), indicating that the observed

variation represents stable individual differences in the ability to produce coherent speech.

Changes in global coherence over the speech production period were also assessed. The mean coherence at each 5 s block is plotted in Supplementary Fig. 1. There was a strong tendency for speech produced later in a response to be less coherent than early speech ($r = -0.93$). This likely occurred because deviations from the prescribed topic tended to occur later in the production period, after the most salient information had been provided. To ensure that relationships between coherence and neural activity were not confounded by this factor, time within the speech production period was included as an additional covariate during neuroimaging analyses (as shown in Fig. 1). Supplementary Fig. 1 also shows the mean number of words produced in each block. Participants tended to produce fewer words in the first 5 s of each response but otherwise there were no systematic changes in speech rate over the 50 s response period (the correlation between time in the response and number of words produced was 0.41 but this fell to 0.05 if the first block was excluded). In other words, there was no evidence that participants slowed their speech rate or ran out of things to say towards the end of the speech periods. There was also no relationship between the number of words produced in a block and the coherence of that block (see section Relationship of coherence with other speech characteristics).

**Activation for speech planning and production.** Regions of significant activation for speech planning and production, relative to automatic speech baselines, are shown in Fig. 2 (for peak co-ordinates, see Supplementary Table 1). Speech planning activated regions typically associated with semantic processing, including left prefrontal cortex, bilateral anterior temporal cortex (specifically the superior temporal sulcus and middle temporal gyrus)

and the angular gyrus. Pronounced medial occipital activation was also observed, presumably related to reading of the written topic prompts. Speech production activated a somewhat similar network of regions, though with a greater left-hemisphere bias. Inferior prefrontal activity was centred on BA45 and activation was also observed in dorsolateral prefrontal cortex and the supplementary motor area, the ventrolateral surface of the anterior

temporal lobe (ATL) and the left angular gyrus. These regions are similar to those reported in previous studies of extended speech production in young adults[15,17,20].

Contrast estimates for speech planning and production in the prefrontal ROIs are shown in Fig. 3b. There appeared to be a strong left-hemisphere bias in the activation of both BA45 and BA47. To test formally for this effect, a $2 \times 2$ ANCOVA

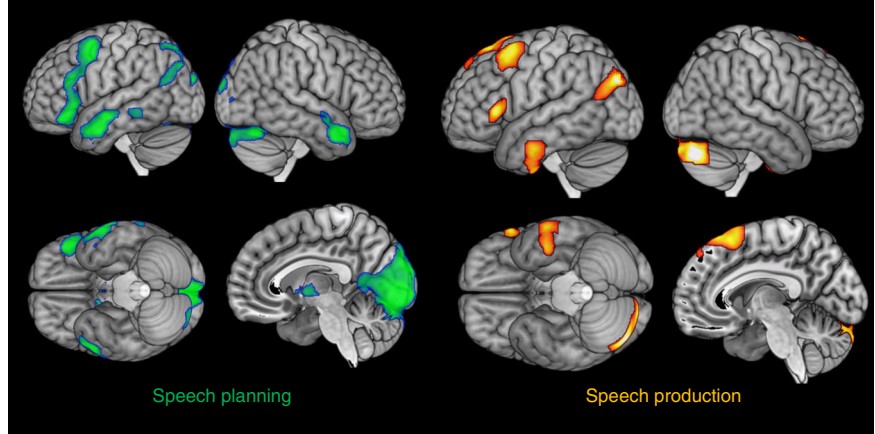

**Fig. 2** Activation for speech planning and production. Planning and production of extended speech were contrasted with the automatic speech baseline

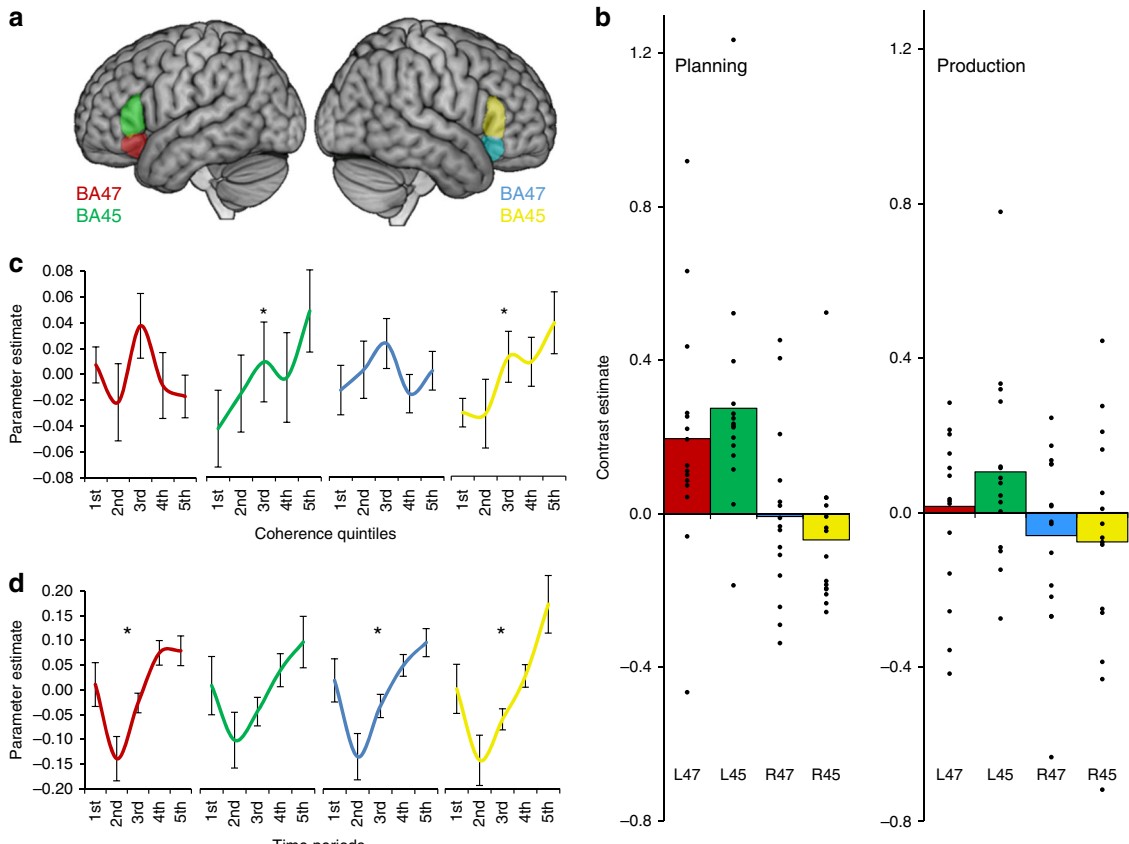

**Fig. 3** Results in prefrontal regions of interest. **a** Anatomical ROIs were specified for left BA47 (red), left BA45 (green), right BA47 (blue) and right BA45 (yellow). **b** Estimates for contrasts of extended speech over automatic speech (points represent individual participants). **c** Effects of coherence on activation in each ROI. Speech blocks were divided into five sets based on their coherence (where 1st quintile = least coherent responses and 5th quintile = most coherent). Estimates for each set are relative to the overall speech-related activation for the region. A significant linear effect of coherence was observed in left and right BA45 (asterisks indicate $p < 0.05$; one-sample $t$-test). **d** Change in activation over time within speech production periods. Speech blocks were divided into five sets based on their temporal position in the speech elicitation period (where 1st = first 10 s and 5th = final 10 s). All regions except left BA45 showed a linear effect of time on activation (asterisks indicate $p < 0.05$; one-sample $t$-test). Bars indicate standard error of the mean

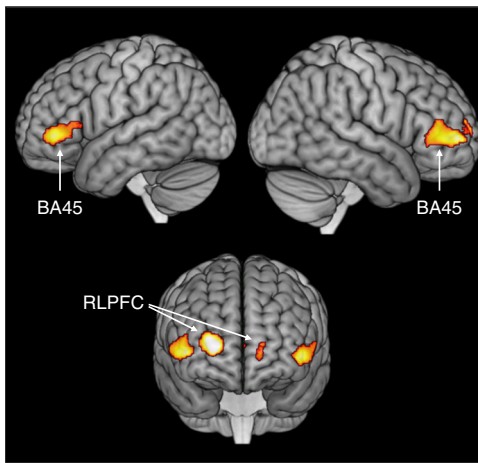

**Fig. 4** Areas of greater activation when participants produced more coherent speech

was performed for each phase of the task, including region (BA45 vs. BA47) and hemisphere (left vs. right) as factors. There were significant effects of hemisphere for planning ($F(1,13) = 40.4$, $p < 0.001$) and production ($F(1,13) = 10.5$, $p = 0.007$), in both cases indicating greater activity in the left hemisphere. There were no main effects of region but there were region×hemisphere interactions for both phases of the task (planning: $F(1,13) = 5.57$, $p = 0.035$; production: $F(1,13) = 5.07$, $p = 0.042$), indicating greater activation of BA45 compared with BA47 in the left hemisphere only.

**Activation varying with coherence**. The second analysis identified regions whose activity was correlated with the global coherence of the speech produced. In the whole-brain analysis, two prefrontal activation clusters showed increased activation when participants produced more coherent speech (see Fig. 4; peaks reported in Supplementary Table 2). The right-hemisphere cluster included BA45 and a region of rostrolateral prefrontal cortex (RLPFC; BA10). Left-hemisphere activation was centred on BA45. No regions exhibited negative effects of coherence.

Effects of coherence in the BA45 and BA47 ROIs are illustrated in Fig. 3c. Left and right BA45 showed a clear linear increase in activity as utterances became more coherent, while no such effect was present in BA47. The effects of coherence in the prefrontal ROIs were analysed with a $2 \times 2$ (hemisphere × region) ANCOVA. This revealed a main effect of region ($F(1,13) = 7.34$, $p = 0.018$), confirming that the effect of coherence was significantly larger in BA45 compared with BA47. There were no between-hemisphere differences in the effect of coherence ($F(1,13) = 0.62$, $p = 0.44$).

To explore the timing of coherence-related activity, two further models were estimated for the neuroimaging data. These were an early model in which coherence values associated with speech were shifted backward in time by 5 s and a late model in which they were shifted forward by 5 s. Significant effects of coherence in these models are shown in Supplementary Fig. 2. The late model principally revealed activation in left BA45, indicating that activation associated with highly coherent speech in this area persisted after the production of the speech itself. However, the early model revealed larger coherence-related activations in RLPFC bilaterally and in right BA45. This indicates that these regions increased activity immediately prior to the production of highly coherent speech, suggesting that they may play an earlier role in shaping the content of upcoming utterances.

**Activation varying over time during speech production**. The third analysis identified regions whose activity increased over time during a speech production act (e.g., regions whose activity was greater at the end of a 50 s production period, relative to its start). Results of the whole-brain analysis are shown in Supplementary Fig. 3 (activation peaks in Supplementary Table 2). A number of regions showed increasing activation as each speech period progressed, including dorsomedial prefrontal cortex, right IFG, bilateral ATL and the right angular gyrus. Reduced activity over time was observed in primary auditory regions.

Change over time in the IFG ROIs are plotted in Fig. 3d. All regions showed broadly similar patterns throughout the speech window, with high levels of activation during the first 10 s of production which then dropped before increasing monotonically over the remainder of the period. One possible interpretation of this pattern is that it reflects high early executive demands when participants initiate their speech act, after which demands subside before gradually building over time as the continued production of speech becomes more challenging. A $2 \times 2$ (hemisphere × region) ANCOVA performed on the effects of time in IFG revealed no main effects, although there was an interaction between hemisphere and region ($F(1,13) = 7.37$, $p = 0.018$). This appeared to arise because there was a more pronounced effect on time in the right hemisphere than in the left, but only within BA45.

**Relationship of coherence with other speech characteristics**. There are a number of other characteristics of speech which could conceivably be correlated with the global coherence measure. For example, a faster speech rate could be associated with a greater propensity to deviate from the prescribed topic, because participants would be more likely when speaking quickly. To ensure that the observed effects of coherence could not be due to such unanticipated confounds, I investigated a number of other properties of speech produced. These comprised: the number of words produced in each block of speech, its type:token ratio, proportion of closed class words and the mean frequency, concreteness, age of acquisition, semantic diversity and phoneme length of nouns produced during speech (for further details of these measures, see Supplementary Methods). An alternative measure of coherence, local coherence, was also computed.

Correlations among these speech measures are reported in Supplementary Table 3. All of the correlations with global coherence were less than 0.2 in magnitude, indicating that was not strongly related to other characteristics of the speech produced (in particular the correlation with number of words produced was −0.01). The only exception was local coherence, which was correlated with the main global coherence measure at 0.39, suggesting that utterances that were strongly related to the main topic of the response (globally coherent) were also unlikely to contain large shifts in subject (locally coherent).

To further explore the structure among the speech measures, I ran a principal components analysis. The results are reported in Supplementary Table 4; they closely replicate the structure found previously in an independent set of speech samples[12]. Four latent factors were identified which appeared to correspond to the following aspects of speech production:

1. Use of complex vocabulary (long, low frequency, late-acquired words)
2. Use of highly specific terms (words low in semantic diversity and high in concreteness)
3. Coherence (both global and local coherence strongly loaded this factor)
4. Verbosity (high number of words and high proportion of closed class words)

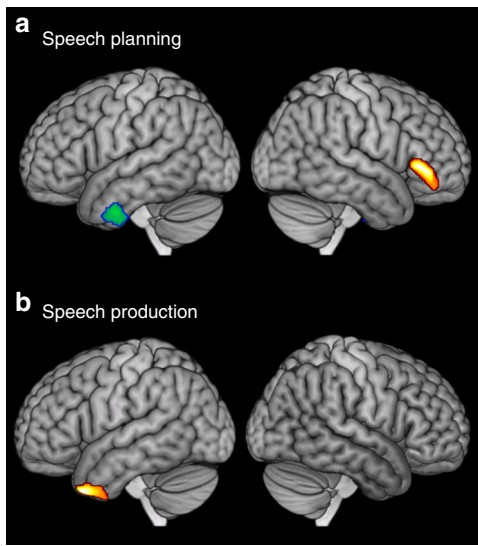

**Fig. 5** Activation during speech planning and production correlated with individual differences in coherence. Red-orange shows regions with greater activation in highly coherent participants and blue-green shows regions with greater activation in less coherent participants. **a** Speech planning; **b** speech production

The principal components analysis provided an opportunity to (a) test that the effects of coherence remained significant when controlling for other characteristics of speech, (b) test the results observed for the global coherence measure generalised to a more general coherence factor that was derived from both global and local coherence. Accordingly, a supplementary neuroimaging analysis was performed in which neural activity was simultaneously predicted from the scores on the four latent factors described above (as well as time within the block). Supplementary Fig. 4 shows areas in which activation was positively correlated with the coherence factor. The pattern closely resembled the main analysis in Fig. 4: significant clusters were again found in left and right BA45 and in RLFPC. This indicates that the effects of coherence are still present when controlling for other factors that characterise speech production, and that they are present when using a latent measure of coherence that is influenced by local, as well as global, coherence.

**Activation differences between more and less coherent people**. Finally, an exploratory analysis was conducted to identify differences in activation between more and less coherent speakers. The mean global coherence level for each participant was included as a covariate in second-level analyses of speech planning and speech production, results of which are shown in Fig. 5 (activation peaks in Supplementary Table 5). Thus, while the previous analysis identified activity that varied with coherence within individuals, this analysis aimed to identify activity that differed across speakers as a function of their coherence. It is important to note that there are a number of other factors that may differ between more and less coherent speakers—for example, educational level, age, general knowledge and level of cognitive function —and these factors were not controlled for. Therefore, while this analysis identifies regions where activation varied between high and low coherence speakers, it is not possible to determine whether these effects are a direct consequence of coherence or are mediated by other factors.

Individual differences were observed during both the planning and production phases of speech. More coherent participants

tended to show greater activation in the right IFG during planning of speech (including portions of BA47 and BA45; see Fig. 5). People who spoke less coherently exhibited greater activity in left ATL during speech planning. This pattern was partially reversed during the speech production period, where a region of left ATL was associated with greater activation in more coherent speakers. No areas showed a negative association with coherence during production.

## Discussion

The ability to maintain coherence while speaking is critical for effective communication but declines in later life. Here, fMRI was used in healthy older people to investigate the neural correlates of coherent speech production. When speaking for 50 s periods about specified topics, older people activated a left-lateralised network of frontal, anterior temporal and inferior parietal regions, similar to that observed in previous studies with young people[15,17,20]. In a major advance on previous studies, formal analyses were performed on speech produced in the scanner to determine its global coherence (i.e., the degree to which utterances were meaningfully related to the subject probed). This allowed discovery of areas in which increased neural activity was correlated with the production of highly coherent utterances. When participants spoke with high coherence, they showed greater activation in two distinct areas of prefrontal cortex: bilateral IFG (BA45) and RLPFC. Both regions are associated with goal-directed executive control of behaviour. BA45 has been implicated specifically in the selection of task-relevant information from semantic knowledge[29,33] while RLPFC contributes to the planning of complex behavioural sequences[40,41]. These findings have major implications for how understanding how people maintain coherence and focus during speech production and, more broadly, for the role of left and right IFG in controlled cognition in later life.

The most notable result in the study is that, as predicted, activity in left and right BA45 was positively correlated with coherence during speech production. In other words, this region showed an increased haemodynamic response when participants produced utterances that were closely related to the current topic, relative to occasions when they deviated off-topic. These results suggest that BA45 plays a central role in regulating the selection of topic-relevant information during speech production (i.e., semantic selection). This conclusion is consistent with evidence for left BA45 involvement in semantic selection across a range of experimental tasks[28,33,42,43]. For example, this region shows increased activation in comprehension when participants are asked to attend to specific semantic features and avoid competition from irrelevant semantic associations[29,34]. In semantic single-word production tasks, BA45 activity is greater for stimuli that prime multiple, competing possible responses[29,44]. The present study demonstrates for the first time that BA45 is also involved in selecting from activated semantic knowledge during extended periods of natural speech.

Why is selection so important for coherent communication? A conversational cue, such as "what's your favourite season?", may automatically cause a wide range of general semantic knowledge to become activated. Some of this information will be useful in answering the question and some less so. Coherent communication requires the speaker to select the subset of information which is directly relevant at the current time, while suppressing aspects of knowledge that have been activated but are less pertinent. Behavioural work supports the idea the effective semantic selection is critical for maintaining coherence in speech. In a recent study, Hoffman et al. found that individuals who were highly skilled at semantic selection (assessed using a forced-

choice "feature selection" task[29]) produced more coherent speech[12]. This effect was independent of the breadth of participants' semantic knowledge (assessed with vocabulary tests) and of non-semantic, domain-general executive function. This study suggests an important role for semantic processes in regulating speech. It is important to note, however, that the knowledge that drives speech production is not solely semantic in nature— memories of specific events and experiences also shape and influence the content of our utterances. Nor is the involvement of BA45 in competition resolution limited to the semantic domain: competition in episodic and working memory tasks also increase activation in this region[45–47]. The neuroimaging literature therefore indicates a general role for BA45 in selecting the most relevant knowledge retrieved from memory, irrespective of the specific type of information.

In contrast, the more anterior portion of IFG, BA47, was not related to speech coherence (and differed significantly from BA45 in this respect). Although left BA47 is also associated with the control and regulation of semantic knowledge, it is thought to serve a different function to BA45[33]. It is typically described as supporting "controlled retrieval" of semantic knowledge when stimulus-driven activation of knowledge is insufficient to complete the task at hand. Under these circumstances, BA47 is thought to support a goal-directed controlled search through the semantic store for less salient information[30,48]. The present results suggest that increased engagement of BA47 does not result in more coherent speech. This may be because the less salient knowledge accessed through controlled retrieval is less centrally related to the topic under discussion, so carries a higher risk of deviation off-topic. Although BA47 showed no relationship with coherence, it did display increasing activity over time during speech production periods (along with BA45). This suggests that controlled retrieval is more engaged during the later stages of a response, once the participant has exhausted the most salient knowledge about the topic at hand and is forced to search for other relevant information.

In our older participants, the association of BA45 activity with high-coherence speech output was present in both hemispheres. In contrast, the majority of studies in young people indicate a dominant role for the left IFG in controlled semantic processing. In younger people, IFG activation during verbal semantic processing is strongly left-lateralised, especially for speech production tasks[49–52]. When right IFG activation is observed during semantic processing, it is less extensive and typically found under conditions of high demand[28,53]. Conversely, older people show a more bilateral pattern of frontal recruitment during semantic processing[35,38,49], which is in line with reduced lateralisation in frontal activity across a variety of domains[37]. The consequences of this age-related reduction in laterality continue to be debated. Some researchers hold that increased recruitment of right IFG serves to compensate for decreased efficiency of processing in left prefrontal regions[54,55]. Others hold that additional right-hemisphere activation is a symptom of reduced specificity in brain activity in old age and does not benefit task performance[38,56].

Although the present study was not designed to adjudicate on these issues directly, our results are more consistent with the compensatory view of right prefrontal activation. Overall, IFG activation during speech planning and production was left-lateralised in our participants. This suggests that, even in older people, the left hemisphere is dominant in guiding production of extended speech. However, the profile of right IFG regions with respect to coherence and time was very similar to that of left IFG: right BA45 displayed increased activity when participant produced highly coherent utterances and right BA45 and BA47 both showed increasing activation over each speech production period. These results suggest that right IFG was engaged in the regulation

of speech content, albeit at lower overall levels of activity than left IFG. In addition, the individual differences analysis revealed that the more coherent individuals in the study exhibited greater right IFG activation during the planning period before beginning to speak. This result should be interpreted with caution, given the limited sample size. It does, however, suggest that greater right IFG activity preceding speech is associated with a more positive behavioural outcome. This association of right prefrontal engagement with improved behavioural performance in older people has been reported previously in other domains but not in speech production[57–59]. However, one previous fMRI study has investigated the relationship between a coherence measure, derived from speech produced out of the scanner, to neural activity during a word recognition task[60]. In their 11 healthy participants (mean age = 47), greater coherence scores were associated with greater activation in right middle frontal and precentral cortex. This study also suggests that more coherent speakers engage right prefrontal regions to a greater degree during language processing, although in that study neural activity was recorded during a receptive task and not during production.

Finally, the only region aside from IFG to show a relationship with coherence was RLPFC. This frontopolar region was more active during production of highly coherent speech, which is in line with the idea that RLPFC is involved in the planning and sequencing of complex, goal-directed behaviours[40,41,61]. In particular, Badre and Nee[62] have proposed that RLPFC controls the sequencing and execution of sub-tasks in service of an over-arching goal, especially when this process is informed by learned knowledge structures (schemas) coded by the ventromedial prefrontal cortex. This organised sequencing of information is critical for producing a well-structured and coherent narrative. Indeed, many of the prompts used to elicit speech here encouraged participants to describe well-learned behavioural schemas (e.g., "What do people do when getting ready for work in the morning?"). The most coherent responses were those that accurately narrated the schema content without deviating to other topics. Interestingly, the time-shifting analysis indicated that RLPFC showed increased activation in the 5 s prior to the production of highly coherent utterances, as well as during production itself. This suggests that this area contributes to coherence prior to production, which is again consistent with hierarchical models of cognitive control which assign a high-level planning role to this area[40,41]. In contrast, the later effects in left BA45 are more consistent with a downstream filtering role, to prevent irrelevant information from intruding into the narrative stream.

Finally, it is important to note that while coherence is often a desirable quality in communication, this is not true in all contexts. For example, if one's aim is to entertain one's interlocutor, it may be advantageous to adopt a less focused and more digressive narrative style, which will make for a more interesting story[6]. Nevertheless, there are many situations in everyday life in which information must be communicated succinctly, and where maintaining focus on the topic at hand is paramount. This investigation of the neural correlates of coherent speech has revealed that multiple prefrontal control systems contribute to the maintenance of coherence. It suggests that timing of these contributions differ, with the most anterior RLPFC region making an earlier contribution than the left IFG. These data will allow future investigations to probe in more detail the precise roles of these systems in speech regulation and their differential recruitment in young vs. older people. This will provide further insights into how complex verbal interactions are planned and regulated.

## Methods

**Participants**. Fifteen healthy older adults were recruited from the psychology department's volunteer panel at the University of Edinburgh. All had previously

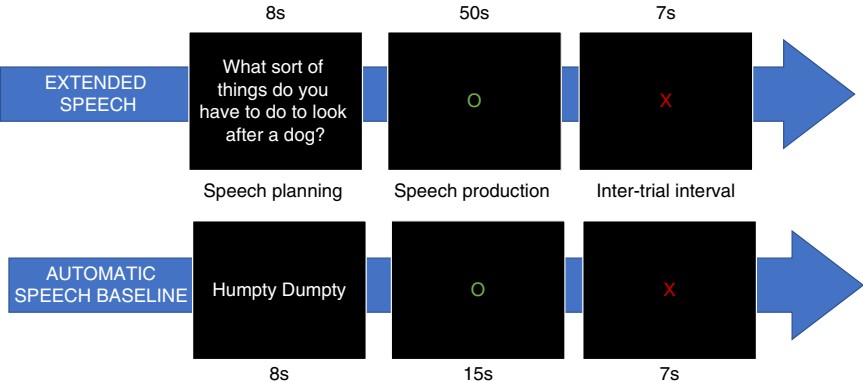

**Fig. 6** Illustration of single trials

taken part in a larger behavioural study investigating cognitive predictors of coherent speech[12,63]. Their mean age was 77.5 (s.d. = 8.4; range = 67–92) and they had completed an average of 15.0 years in education (s.d. = 2.9; range = 10–20). All reported to be in good health with no history of neurological or psychiatric illness. The Mini-Mental State Examination[64] was used to screen for possible cognitive impairments; all participants scored at least 28/30. Their scores on a range of neuropsychological tests are provided in Supplementary Table 6. Informed consent was obtained from all participants, the study was approved by the University of Edinburgh Psychology Research Ethics Committee (1-1516/1) and all relevant ethical regulations were complied with.

**Design and procedure**. The study contrasted extended speech production with a baseline consisting of automatic speech (reciting a nursery rhyme). The structure of trials is shown in Fig. 6. During the extended speech task, participants were asked to speak for periods of 50 s at a time in response to a series of prompts. There were 20 prompts, which were designed to probe particular a range of semantic knowledge (e.g., "What sort of things do you have to do to look after a dog?"; for a full list, see Supplementary Methods). Fourteen of the prompts were taken from a previous behavioural study[12]. Each trial began with a speech planning phase lasting 8 s, in which the written prompt was presented on screen. Participants were instructed to prepare to speak about the subject given but not to begin speaking until the prompt was replaced by a green circle. They were asked to speak about the prompted subject for 50 s, at which point the green circle would be replaced by a red cross. They were then instructed to stop speaking and wait for the next trial, which would begin after 7 s. The structure of the baseline task was similar, except that the prompt was always "Humpty Dumpty". They were asked to begin reciting this nursery rhyme upon seeing the green circle. To reduce participant fatigue, baseline trials lasted for only 15 s. If participants reached the end of the nursery rhyme before the 15 s had elapsed, they were asked to start again from the beginning. The baseline condition therefore involved production of grammatically well-formed continuous speech, but without the requirement to generate novel, meaningful utterances.

Functional imaging took place over two runs of approximately 16 min each, with each run including ten trials of each type in a pseudo-random order. The order in which prompts were presented was counterbalanced over participants. Prior to scanning, participants were presented with the words to Humpty Dumpty and given an opportunity to practice both tasks.

**Processing of speech samples**. Data analysis steps are illustrated in Fig. 1. Speech was recorded using an MR-compatible microphone and audio files processed with noise cancellation software to reduce scanner noise[65]. Responses to each prompt were transcribed and measures of global coherence were obtained by subjecting speech samples to computational analyses based on latent semantic analysis (LSA)[66]. LSA provides the user with vector-based representations of the meanings of words, which can be combined linearly to represent the meanings of whole passages of speech or text. Using related methods to other researchers[67,68], I used these representations to characterise the content of each speech sample, defining global coherence as the similarity of a sample's semantic content to the prototypical semantics typically produced in response to the same prompt. Global coherence calculated in this way has high internal reliability and test-retest reliability and is highly correlated with human ratings of coherence[12].

Global coherence was computed using methods first described by Hoffman et al.[12]. Analyses were implemented in R; the code is publicly available and can easily be applied to new samples (https://osf.io/8atfn/). The computation process is illustrated in Fig. 7. For a response to a given prompt, coherence was calculated as follows. First, an LSA representation was computed for all other participants' responses to the same prompt, excluding the response currently under analysis. These were averaged to give a composite vector that represented the typical semantic content that people produced when responding to the prompt (further details of the LSA space and averaging procedures are provided as Supplementary

Methods). For example, the composite vector for the prompt "Which is your favourite season?" would be close to the vectors for summer, autumn, weather, sunshine and so on, as these concepts were frequently used in responses to this prompt.

Next, the target response was analysed using a moving window approach. For each word in the response, a 20-word window was created consisting of the current word and the 19 words that preceded it. An LSA vector was computed for the window: this provided a vector representation of the semantic content of the speech produced in the immediate run-up to the current word. This vector was then compared to the composite prototypical vector using a cosine similarity metric (the cosine gives a value for similarity between zero and one; throughout the paper, these values were multiplied by 100 for ease of presentation). The result of this calculation was therefore a value, assigned to the final word in the window, that indicated how similar the speech produced in the current window was to the typical semantic content of responses to that prompt. A high coherence value indicated that the speech was closely related to the topic being probed. A low coherence value indicated that the participant was speaking about topics that were semantically unrelated to the topic being probed. Thus, the measure captured the degree to which participants maintained their focus on the topic about which they were asked, in line with the definition of global coherence used by other researchers[3,7]. Examples of high and low coherence responses are shown in Supplementary Table 7. Although not of primary interest in the present study, a measure of local coherence was also computed using LSA and included in supplementary analyses (see Supplementary Methods for details).

To obtain a dynamic measure of coherence at each stage of speech production, each 50 s speech production period was divided into shorter blocks of 5 s. Coherence in each block was calculated by averaging the coherence values associated with each of the words produced in the block. Reliable coherence estimates could not be obtained for the first block in each response, due to a lack of prior speech to analyse. In these cases, I used the coherence value for the second block instead. A block length of 5 s was chosen to model coherence for two reasons. First, changes in global coherence tend to occur over relatively long timescales—a shift in discourse away from its original topic typically occurs over the course of at least one or two sentences. Given that participants produced an average of 10 words every 5 s, this level of temporal resolution seemed appropriate to capture variations in the topic of speech. Second, a block length of 5 s is well matched to the temporal resolution of the BOLD signal indeed, fMRI can be sensitive to variation in linguistic content at considerably shorter time-scales than this; e.g.,[69]. Other techniques such as EEG and MEG afford much greater temporal resolution, of course, and have provided valuable insights into coherence building in comprehension[70,71]. However, such techniques cannot be used during extended speech production.

**Measurement of other characteristics of speech**. In addition to the global coherence measure described above, a number of other speech markers were computed. These comprised: the number of words produced during each 5 s block, a measure of local coherence, type:token ratio, proportion of closed class words and the mean frequency, concreteness, age of acquisition, semantic diversity and phoneme length of nouns produced during speech (for further details of these measures, see Supplementary Methods). I computed the correlation between each of these additional measures and the primary coherence measure, to test whether any strongly covaried with coherence. To further explore the structure among speech characteristics, a principal components analysis was performed on all measures, including global coherence, which resulted in the extraction of four latent factors (which were the only factors with eigenvariates greater than one and together explained 66% of the variance within the set of measures). The factors were promax rotated to aid interpretation. Scores on these latent speech factors were later used as predictors of neural activity in a supplementary analysis (see below).

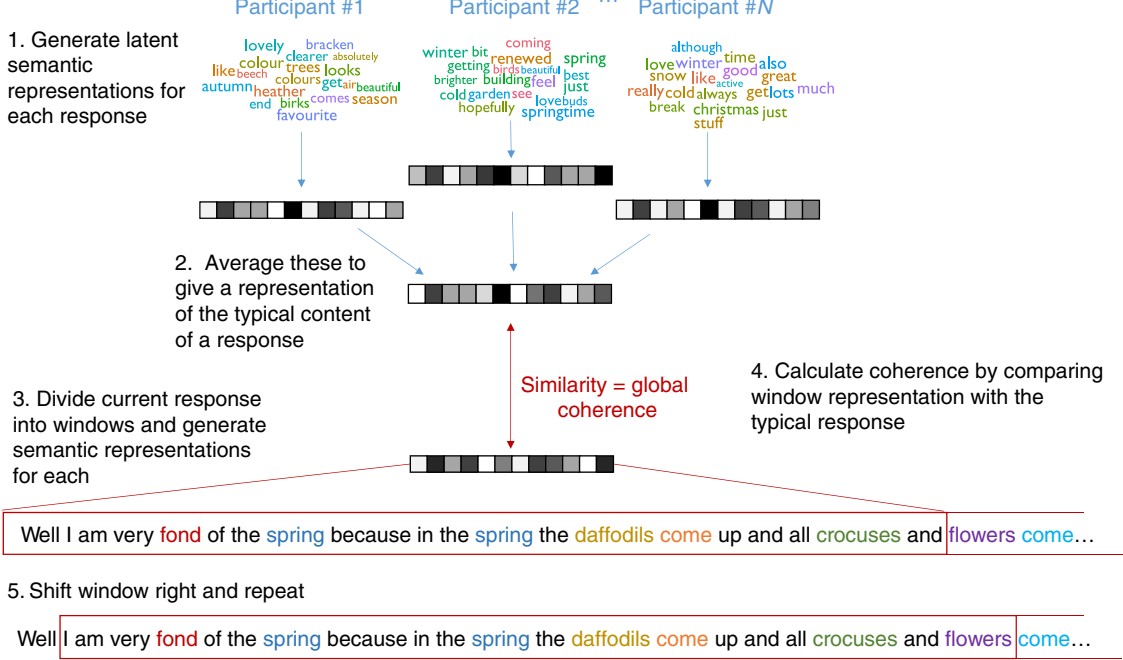

**Fig. 7** Process for computing coherence in speech samples. 1. LSA was used to generate a vector coding the semantic content of each participant's response to a given prompt. 2. These were averaged to give a representation of the semantics of a prototypical response. 3. The current response was divided into windows of 20 words and semantic representations computed for each window. 4. Global coherence was computed by calculating the similarity of the speech produced in the window to the prototypical response to the prompt. A high level of similarity indicated that the current response was closely related to the material typically associated with the topic. This coherence value was assigned to the final word in the window. 5. The window was shifted one word to the right and the process repeated

**Image acquisition and processing**. Images were acquired on a 1.5 T GE Signa Horizon scanner with an eight-channel head coil. The EPI sequence included 30 slices covering the whole brain with TE = 40 ms, TR = 2200 ms, flip angle = 90°, 64 × 64 matrix, reconstructed in-plane resolution = 3.75 × 3.75 mm, slice thickness = 4.0 mm. 900 volumes were acquired over two runs. T1 and T2 structural images were acquired for co-registration. An MP RAGE sequence was used to acquire a whole-brain image with 256 × 256 matrix, in-plane resolution = 0.9375 mm, slice thickness = 1.3 mm. A T2 image was acquired with 256 × 256 matrix, in-plane resolution = 0.9375 mm, slice thickness = 2 mm.

Images were analysed in SPM12. Functional image acquisition during overt speech production can be challenging due to additional head movement. A number of steps were taken to minimise the effects of movement on image quality. First, following spatial realignment, images were processed using the FIACH toolbox which is designed to ameliorate effects of head motion in studies that employ overt speech production[19]. The toolbox has two functions. First, it identifies and removes signal spikes in the time-series of individual voxels. Second, it uses principal components analysis to identify noise components across the whole brain in each participant's data. These noise components can be included as covariates of no interest in later analyses for a similar approach, see ref. [20]. Here, the first six noise components were included as covariates in all first-level analyses, in addition to the six movement parameters obtained during spatial realignment. Finally, a measure of each participant's propensity to move in the scanner was obtained as follows. The mean scan-to-scan change was calculated for each of the six movement parameters and these were averaged to give a mean scan-to-scan displacement value for each participant for a similar approach, see ref. [72]. This value was included as a nuisance covariate in all second-level analyses.

Functional images were then normalised to MNI space, resampled to 2 mm isotropic voxel size and smoothed with a 12 mm FWHM Gaussian kernel. Following pre-processing, data were treated with a high-pass filter with a cut-off of 180 s and analysed using a general linear model. Four event types were modelled: extended speech planning, extended speech production, automatic speech planning and automatic speech production. Speech planning periods were modelled as 8 s blocks. The modelling of speech production periods is illustrated in Fig. 1. Each speech production period was modelled as a series of concatenated 5 s blocks. This allowed specification of two parametric modulators that coded the characteristics of the speech produced in each 5 s block. The first of these was the coherence of the speech produced, calculated as described earlier, and the second was the time point within the 50 s speech production period (see Fig. 1). Blocks were modelled as boxcars convolved with the canonical hemodynamic response function. Prior to entry in the model, the parametric regressors for each scanning run were mean-centred and values more than two standard deviations from the mean of the run were winsorized.

In addition, three supplementary first-level models were estimated. The first two models assessed the temporal characteristics of coherence-related activation. These comprised an early model in which coherence values were shifted back in time by 5 s and a late model in which they were shifted forwards (e.g., the coherence value for words produced in block 4 of would be assigned to block 3 in the early model and to block 5 in the late model). These models tested for activation that either preceded or followed the production of high/low coherence speech.

The final model tested whether the results still held when controlling for other speech characteristics with which coherence may be correlated. This model made use of the latent factors identified through principal components analysis of the full set of speech characteristics described earlier. In this model, there were therefore five parameters for each block of speech: its score on each of the four latent factors plus its time within the speech production period.

**Analyses**. Analyses were performed across the whole brain and in anatomically-defined prefrontal ROIs. A voxel-height threshold of $p < 0.005$ was adopted for whole-brain analyses (all second-level analyses were one-sample $t$-tests), with correction for multiple comparisons performed at the cluster level. The minimum cluster size was determined using a Monte Carlo analysis[73]. This modelled the entire smoothed image volume, assumed an individual voxel type-1 error rate of 0.005 and ran 5000 simulations to determine the minimum cluster size associated with a corrected $p < 0.05$.

The first set of analyses considered activation for speech planning (extended speech planning minus automatic speech planning) and production (extended speech planning minus automatic speech planning). Subsequent analyses tested for effects of coherence and time on neural activity during speech production. Finally, an exploratory individual differences analysis was performed, to determine whether activity differed between highly coherent and less coherent speakers. For this analysis, the mean level of coherence for each participant's speech was calculated over all of their responses and these values were added as a covariate to the second level analysis.

**Prefrontal regions of interest**. Anatomical ROIs were specified a priori in the left and right IFG (see Fig. 3a), using the Harvard-Oxford probabilistic atlas[74]. The BA45 region was defined as voxels with >30% probability of falling in the pars triangularis region and BA47 as those with >30% probability of falling in pars orbitalis. The pars orbitalis region includes areas of medial prefrontal cortex that have not been linked with language processing. To exclude these, voxels more medial than $x = \pm 30$ mm were removed from the mask.

The Marsbar toolbox[75] was used to extract parameter and contrast estimates for each participant. In addition, to aid visualisation of the parametric effects of coherence and time, the rfxplot package[76] was used to estimate effects at five levels of each parameter.

**Code availability**. The R code used to compute coherence for speech samples is available at https://osf.io/8atfn/.

## Data availability
The neuroimaging data analysed during the current study are available in the Open Science Foundation repository, https://osf.io/9ca5g/ (DOI: 0.17605/OSF.IO/9CA5G).

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

## Acknowledgements

The study was funded by the University of Edinburgh Centre for Cognitive Ageing and Cognitive Epidemiology, part of the cross council Lifelong Health and Wellbeing Initiative (MR/K026992/1). Funding from the Biotechnology and Biological Sciences Research Council (BBSRC) and Medical Research Council (MRC) is gratefully acknowledged. Imaging was carried out at the Edinburgh Imaging Facility (www.ed.ac.uk/edinburgh-imaging), University of Edinburgh, which is part of the SINAPSE collaboration (www.sinapse.ac.uk). I am grateful to Cyril Pernet and Sergio Della Sala for advice on study design and participant screening.

## Additional information

**Competing interests:** The author declares no competing interests.

