## [Peer Review File · Nature Communications]

Reviewers' comments:

Reviewer #1 (Remarks to the Author):

The importance of the topic is introduced well, with convincing statements on how changes in the coherence of speech can reduce the effectiveness of communication and be associated with higher levels of stress. The manuscript is written in an elegant manner and thus is enjoyable to read. The author makes a compelling case for coherence being a critical part of communication and that a reduction in coherence should be measurable and have a pattern of brain function that is different from that associated with speech that is more coherent.

Nonetheless, this manuscript leaves me with more questions than answers, of which the two most fundamental concern themselves with (1) how to suitably define coherence in speech and (2) whether this extremely rapid dynamical process in speech is possible to measure using functional MRI where the hemodynamic response that is measured is magnitudes slower. It is very possible that my concerns are not justified but the onus is on the author to persuade a skeptical reader. I detail my thoughts sequentially as I read through the manuscript:

Abstract & Introduction

-The statement that coherence declines with age needs to be qualified as it is a blanket statement that is rather ambiguous (i.e., what age is it this apparently occurs; is it topic-specific; is it state-related; is it a global decline or something that occurs in some conversations? etc).

- The natural language processing methods (LSA) that are introduced are no longer 'novel' as the author implies. They were introduced decades ago and indeed the paper the author cites is from 2 decades ago (Landauer & Dumais, 1997).

-The statement that no study has 'attempted to relate neural activity recorded during spontaneous speech to its coherence' is a bold one, and as worded may be true. Nonetheless, the author neglects to talk of other methods that have been applied which are also relevant. For example, Tagamets et al (2014) examined the relationship of somewhat similarly computationally derived (LSA) coherence scores from free speech generated outside of the scanner to activation during a word monitoring task performed inside the scanner. The study was conducted in a clinical group (patients with schizophrenia) who often are reported to produce speech that is less coherent than healthy comparison participants, and as such this study seems highly relevant. Of course, the limitation of this particular study was that these two correlated measures were acquired at two different time points and using two different tasks (free speech versus task performance on a simple cognitive task). However, given that the producing or creating coherent speech is the core topic of interest in this manuscript it was surprising that the author did not mention recent research on the spatiotemporal dynamics of meaning construction using technologies with high temporal resolution such as MEG (Bemis and Pylkkänen, 2012; Pylkkänen et al., 2014). There is also a large literature on event-related potentials (ERP) that seems relevant (e.g., Ditman and Kuperberg, 2010). My point is that currently there is a very limited neurobiological understanding of how various aspects of language are integrated, notably how words are combined and meaning is created, but it is hard to reject the notion that temporal resolution is key, yet fMRI is not especially well-suited to examine temporal resolution.

Method

p.6 (and elsewhere) is reference to Hoffman et al. (in press) and in several places it is not clear how the current manuscript differs from this 'in press' manuscript. Also on p.8 the author reverts to 'we' yet the current manuscript is by one author only. Put bluntly, how much overlap is there of the 'in press' article with the current manuscript?

p.8 . The natural language processing methods employed are relatively sophisticated (but presented minimalistically and refer the reader to the 'in press' article), and it is very positive that the authors use the Open Science Framework platform for sharing their methods. By providing the code for calculating the local and global coherence there should be no ambiguity around the

operationalization of the main metrics and this should facilitate replication or comparison of the results of different computational approaches. Nonetheless, it would be helpful if the author is explicit about the semantic representational space that he has employed such that this is easy to establish as this will be critical for the aforementioned issues (i.e., subtle differences in computational platforms can have big effects). Furthermore, the author needs to help the reader understand how the coherence numbers in Figure 2 (panels 2 and 3) were generated (i.e., what was compared to what) rather than having to figure it out from reading the code on the shared platform. Also, it is confusing that the author mentions that he created coherence values on a scale of 0-100 (p.12 footnote) yet in all the tables and figures (and in the original papers by for example Landauer and Dumais (1997)) cosine values ranging from 0-1 are standardly employed. The author needs to clarify this, and the Method referred to in footnote 1 (p.12) does not describe this.

At a conceptual level, after reading this manuscript it is still not clear (i) what coherence means in this study, and (ii) what an increased hemodynamic response equates to?

It seems as though responses are scored as coherent if people say something similar to what the other participants have said on the matter (e.g., if people describe mornings, a good score will not be achieved if one has an unusual morning ritual). This would explain the decrease in "coherence" over the time-windows: people run out of common things to say after a few tens of seconds, like "getting out of bed", "taking as shower", "making coffee", etc. Staying on topic (i.e., saying what the others said) seems to be related to spending more resources (increased blood-flow) in the areas of interest. Or, people who have increased blood flow in these areas relative to when they say the humpty-dumpty verse, seem to be able to create more on-topic speech. So, is the current operationalization "coherence" appropriate? Well, it is one definition (amongst many that are possible), along the lines of a "staying-on-topic global coherence" measure and thus it would be interesting to see how the results would change if a different operationalization of coherence was used such as a "local coherence" metric (that arguably is more intuitively relevant) of how much the topics are "bouncing around" rather than what was used here. In general, the author needs to discuss how 'measure- specific' the current findings might be, namely to what extent they might be affected by subtle nuances in the natural language approach that has been adopted.

Results:

Another caveat to functional neuroimaging is that the results are based on the premise of cognitive subtraction such that the measured activation is a representation of the relative differences in brain activity between two or more brain states elicited by the task and therefore dependent on the intricacies of not only the task of interest but also the "baseline" conditions. Given the centrality of this issue, the onus is on the author to persuade readers that this 'subtraction' has been conducted appropriately. Specifically, regarding the temporal dynamics of coherence (i.e., coherence decreasing over time), could the 5s blocks of speech production be sensitive to pauses (e.g., containing no words, one word, etc.)? One could imagine that there are more pauses and fewer words per speech window at the end of the interval, creating a more labile estimate of coherence, and that this could be a factor in the clear trend of the impressive Supp.Fig.1. The author also mentions this factor on page 16, "Reduced activity over time was observed in primary auditory regions, perhaps due to a slowing of speech production later in the period.". Even if this would probably not invalidate the overall results, it could be interesting to know the magnitude of this slowing, perhaps with an average of number of words produced per speech window.

p.17 – Figure 6 (and elsewhere) – the author refers to an activation pattern in 'highly coherent participants' versus 'less coherent participants'. Is it something specific to participants or to that specific speech sample? Put differently, is this effect that is documented consistent within the individual across testing times? Surely the latter must be demonstrated for the premise of the current manuscript to be valid as otherwise the phenomena being reported are more reflective of a more spurious issue associated with the stimuli used to generate the speech or some transient state-like phenomena...? Either way, this needs to be clarified given the premise of the study that speech coherence declines with age (and thus presumably this is what is intended to be

measured).

Minor points:

P5, line3: It would perhaps be more clear to state: "The main prediction of the current study was that...", or something similar, so the reader knows we have moved past the background information.

P5, line 14: "A recent study indicates...". Perhaps using past tense?

P8, and for results: It may be worth pointing out more clearly where the 20-word window approach was used, and where the 5s windows were used (approx. 10 words and below?). One can assume that the 20-word window was for the more overall measures for each response in total.

P9, Figure 2, panel 3: How do we interpret the y-axis? Hemodynamic response?

Additional comment on panel labels: Figure 4 is labeled with panels A-D, in Figure 2 the panels are labeled with 1-4 (with panel 3 legend referencing an A-C not present in the figure. This may perhaps be uniform between figures for clarity.

P12, line 6: What was the method used to assess test-retest reliability, some variant of the ICC? Would be interesting to see the numbers, e.g., in a bracket after the sentence, with perhaps the second most reliable measure for reference.

P21, third last line: Is "first investigation" perhaps a bit strong statement?

Reviewer #2 (Remarks to the Author):

This is a very interesting study. The methods are clearly described. The statistics are appropriate. The results are clear and the discussion is well structured. I have only a few issues (mostly related to the introduction section and, partially, the discussion). Overall, I think that the paper needs a minor revision.

Major issues:

1. Please provide a theoretical model for the statements at the beginning of the Introduction (e.g., the Framework Building Model by Gernsbacher, 1990 or Levelt's model [1999]). It would be useful to appropriately introduce the hypotheses at the end of the Introduction and interpret the findings from this study with such psycholinguistic models in the Discussion.
2. Before introducing the statement that "more coherent individuals perform better on tests of cognitive and executive control" it would be important to highlight that according to Miyake et al. (2000) and Mozeiko et al. (2011) executive functions are likely involved in different stages of message production (especially in the phases of message planning and framework laying).
3. Still in the Introduction, page 3: "Left IFG has been identified as the key node in a network of regions that control the retrieval and selection of semantic information". In support of this claim, here the authors should also mention that an rTMS study by Marini and Urgesi (2012) suggests that the dorsal aspect of the left IFG is involved in the process of extraction of informative words during a discourse generation task. The inhibition of this area reduced the amount of informative words by increasing the production of off-topic utterances.

References:

Gernsbacher, M. A. (1990). *Language comprehension as structure building*. Hillsdale, NJ: Erlbaum.

Levelt, W. J. M., Roelofs, A., & Meyer, A. S. (1999). A theory of lexical access in speech production. *Behavioral and Brain Sciences*, 22, 1–38.

Marini, A., & Urgesi, C. (2012). Please Get to the Point! A Cortical Correlate of Linguistic Informativeness. *Journal of Cognitive Neuroscience*, 24(11), 2211–2222.

Miyake, A., Friedman, N. P., Emerson, M. J., Witzki, A. H., & Howerter, A. (2000). The unity and diversity of executive functions and their contributions to complex “Frontal Lobe” tasks: A latent variable analysis. *Cognitive Psychology*, 41, 49-100.

Mozeiko, J., Le, K., Coelho, C., Krueger, F., & Grafman, J. (2011). The relationship of story grammar and executive function following TBI. *Aphasiology*, 25, 826-835.

Minor points:

1. Introduction, page 1: “coherence, the degree to each utterance relates to the topic under discussion” here something is missing. Please check.
2. Methods: Supplementary Table 1 – please provide the complete names (not just the acronyms) of the used tests in a legend. Also, please provide more information about the possibility that some individuals were under the age-related cut-off for normality. In case they were all over such cut-off just state it in the body of the text.

Reviewer #3 (Remarks to the Author):

The manuscript presents a novel study investigating the effects of speech coherence on neural activation measured with fMRI.

The study topic is novel and the manuscript well written and would be of interest to the audience of *Nature Communications* and it is great that the data is made available and methods presented very openly. However, I am not convinced that the measure of coherence (presented on its own as currently) is very informative. If I understand the ms correctly, the measure of coherence relies on similarity between an individual's utterances with the group (excluding the individual) composite scores. This seems like a very restricted definition of coherence. My concerns include the following: in at least one other behavioral paper by the author other measures of coherence are also presented (e.g., local coherence, based on the individual's own speech, I think) but not presented here; a range of other definitions of coherence could be arrived at, e.g., based on the whole corpus for each individual or based on similarity to clusters within the other subjects rather than the mean composite from the relatively small group or using methods other than LSA. It is also not clear that semantic coherence is driving the effects seen: e.g., what about differences in the number of syllables/utterances made, or speech errors or speed of speech produced, or level of semantic complexity or size of vocabulary used; equally, what about non-speech related issues such as effects of fatigue, background knowledge/life experiences etc.

The manuscript would be greatly improved if other factors could be accounted for as much as possible. E.g., speech rate/semantic/syntactic errors/repetition as well presenting converging evidence from different definitions of coherence, including those derived from an individual's own language as much as possible.

Reviewer #1:

The importance of the topic is introduced well, with convincing statements on how changes in the coherence of speech can reduce the effectiveness of communication and be associated with higher levels of stress. The manuscript is written in an elegant manner and thus is enjoyable to read. The author makes a compelling case for coherence being a critical part of communication and that a reduction in coherence should be measurable and have a pattern of brain function that is different from that associated with speech that is more coherent.

Nonetheless, this manuscript leaves me with more questions than answers, of which the two most fundamental concern themselves with (1) how to suitably define coherence in speech and (2) whether this extremely rapid dynamical process in speech is possible to measure using functional MRI where the hemodynamic response that is measured is magnitudes slower. It is very possible that my concerns are not justified but the onus is on the author to persuade a skeptical reader. I detail my thoughts sequentially as I read through the manuscript:

I am grateful for the reviewer's encouraging comments and for their helpful questions. I respond to their specific concerns below.

Abstract & Introduction

-The statement that coherence declines with age needs to be qualified as it is a blanket statement that is rather ambiguous (i.e., what age is it this apparently occurs; is it topic-specific; is it state-related; is it a global decline or something that occurs in some conversations? etc).

I have added information to the first paragraph of the Introduction as suggested. It now reads:

"In order to speak coherently on a topic, individuals must constantly regulate their speech output to ensure that they produce statements that are informative and relevant to the topic under discussion. Numerous studies have found that this ability declines in later life, with changes evident in people aged 60 and above (Kintz, Fergadiotis, & Wright, 2016; Marini, Boewe, Caltagirone, & Carlomagno, 2005). Older adults are more likely to produce tangential, off-topic utterances during conversation (Arbuckle & Gold, 1993; Glosser & Deser, 1992) and to provide irrelevant information when telling a story (Juncos-Rabadan, Pereiro, & Rodriguez, 2005; Marini et al., 2005) or describing an object (Long, Horton, Rohde, & Sorace, 2018). Although these effects have been observed in a range of tasks, age-related coherence declines are most pronounced when individuals provide information from their own memory or personal experience, and less severe when they describe visually-presented stimuli, such as pictures or comic strips (James, Burke, Austin, & Hulme, 1998; Kintz et al., 2016; Wright, Koutsoftas, Capilouto, & Fergadiotis, 2014)."

- The natural language processing methods (LSA) that are introduced are no longer 'novel' as the author implies. They were introduced decades ago and

indeed the paper the author cites is from 2 decades ago (Landauer & Dumais, 1997).

“Novel” in this context was intended to refer to the application of LSA methods to quantify coherence, rather LSA itself. I apologise if this was not clear. To avoid confusion, I no longer use “novel” in this context.

-The statement that no study has ‘attempted to relate neural activity recorded during spontaneous speech to its coherence’ is a bold one, and as worded may be true. Nonetheless, the author neglects to talk of other methods that have been applied which are also relevant. For example, Tagamets et al (2014) examined the relationship of somewhat similarly computationally derived (LSA) coherence scores from free speech generated outside of the scanner to activation during a word monitoring task performed inside the scanner. The study was conducted in a clinical group (patients with schizophrenia) who often are reported to produce speech that is less coherent than healthy comparison participants, and as such this study seems highly relevant. Of course, the limitation of this particular study was that these two correlated measures were acquired at two different time points and using two different tasks (free speech versus task performance on a simple cognitive task).

Thanks for bringing this interesting study to my attention. I have added some discussion of this study to the General Discussion where I consider the results of the individual differences analysis. The new text is as follows:

“one previous fMRI study has investigated the relationship between a coherence measure, derived from speech produced out of the scanner, to neural activity during a word recognition task (Tagamets, Cortes, Griego, & Elvevåg, 2014). In their 11 healthy participants (mean age = 47), greater coherence scores were associated with greater activation in right middle frontal and precentral cortex. This study also suggests that more coherent speakers engage right prefrontal regions to a greater degree during language processing, although in that study neural activity was recorded during a receptive task and not during production.”

However, given that the producing or creating coherent speech is the core topic of interest in this manuscript it was surprising that the author did not mention recent research on the spatiotemporal dynamics of meaning construction using technologies with high temporal resolution such as MEG (Bemis and Pylkkänen, 2012; Pylkkänen et al., 2014). There is also a large literature on event-related potentials (ERP) that seems relevant (e.g., Ditman and Kuperberg, 2010). My point is that currently there is a very limited neurobiological understanding of how various aspects of language are integrated, notably how words are combined and meaning is created, but it is hard to reject the notion that temporal resolution is key, yet fMRI is not especially well-suited to examine temporal resolution.

I have added the following text to the Method in response to this comment:

“To obtain a dynamic measure of coherence at each stage of speech production, each 50s speech production periods was divided into blocks of 5s....A block length of 5s was chosen to model coherence for two reasons. First, changes in global coherence tend to occur over relatively long timescales – a shift in discourse away from its original topic typically occurs over the course of at least one or two sentences. Given that participants produced an average of 10 words every 5s, this level of temporal resolution seemed appropriate to capture variations in the topic of speech. Second, a block length of 5s is well-matched to the temporal resolution of the BOLD signal (indeed, fMRI can be sensitive to variation in linguistic content at considerably shorter time-scales than this; e.g., Lerner, Honey, Silbert, & Hasson, 2011). Other techniques such as EEG and MEG afford much greater temporal resolution, of course, and have provided valuable insights into coherence building in comprehension (Bemis & Pykkänen, 2012; Ditman & Kuperberg, 2010). However, such techniques cannot be used during extended speech production.”

Method

p.6 (and elsewhere) is reference to Hoffman et al. (in press) and in several places it is not clear how the current manuscript differs from this 'in press' manuscript. Also on p.8 the author reverts to 'we' yet the current manuscript is by one author only. Put bluntly, how much overlap is there of the 'in press' article with the current manuscript?

Sorry if this was not clear. The “in press” study (which is no longer in press and is now referred to in the ms as Hoffman et al. (2018a)) was a behavioural study in which we investigated which cognitive test scores predict coherence in young and older people. There were no neuroimaging data in this previous study.

Some of the participants from Hoffman et al. (2018a) subsequently took part in the present study. In addition, similar methods were used for eliciting speech samples and computing coherence in the two studies. But the data presented here are new and have not been reported previously.

I apologise for any confusion with the pronouns. Hoffman et al. (2018a) was written with two co-authors; the present study is a solo effort but I think a few “we”s slipped through force of habit. This has now been corrected.

p.8 . The natural language processing methods employed are relatively sophisticated (but presented minimalistically and refer the reader to the 'in press' article), and it is very positive that the authors use the Open Science Framework platform for sharing their methods. By providing the code for calculating the local and global coherence there should be no ambiguity around the operationalization of the main metrics and this should facilitate replication or comparison of the results of different computational approaches. Nonetheless, it would be helpful if the author is explicit about the semantic representational space that he has employed such that this is easy to establish as this will be critical for the

aforementioned issues (i.e., subtle differences in computational platforms can have big effects).

In response to this comment, I have added more details on the computation of coherence. I have placed this information in the Supplementary Materials due to length limitations in the main text.

Furthermore, the author needs to help the reader understand how the coherence numbers in Figure 2 (panels 2 and 3) were generated (i.e., what was compared to what) rather than having to figure it out from reading the code on the shared platform.

As well as the additional information provided in the text, I have added a figure that illustrates the computation process in more detail (Figure 7).

Also, it is confusing that the author mentions that he created coherence values on a scale of 0-100 (p.12 footnote) yet in all the tables and figures (and in the original papers by for example Landauer and Dumais (1997)) cosine values ranging from 0-1 are standardly employed. The author needs to clarify this, and the Method referred to in footnote 1 (p.12) does not describe this.

I apologise for the confusion. The cosine values were multiplied by 100 to place them on a more easily interpretable scale (this was done mainly to be consistent with Hoffman et al. (2018a)). This information has been added to the Method and the footnote edited as follows to be more informative: "As described in Method, the coherence measure quantifies the strength of the semantic relationship between the speech produced and the typical responses made to the same prompt. It uses a cosine similarity metric, which varies between 0 and 1 and is multiplied by 100 here for ease of presentation. 0 therefore indicates speech that has no semantic relationship with the topic being probed and 100 indicates speech that is identical to it."

At a conceptual level, after reading this manuscript it is still not clear (i) what coherence means in this study

To make this clearer, I have expanded the definitions of global and local coherence provided in the Introduction: "Researchers have often made a distinction between local coherence, the degree to which adjoining utterances related meaningfully to one another, and global coherence, the degree to which each utterance relates to the topic under discussion (Glosser & Deser, 1992; Kintsch & Vandijk, 1978). Speech that is tangential or off-topic is therefore said to be low in global coherence, while speech that shifts abruptly between subjects is low in local coherence (although these measures are typically correlated). As most studies have reported that global coherence declines more severely than local coherence in later life (Glosser & Deser, 1992; Kemper, Schmalzried, Hoffman, & Herman, 2010; Marini et al., 2005; Wright et al., 2014), this aspect of coherence was the focus of the present work."

I have also made it clearer throughout the paper that the main coherence measure is a measure of global coherence.

and (ii) what an increased hemodynamic response equates to?

In common with most fMRI research, I assume that an increased haemodynamic response in a region indicates greater engagement of the cognitive processes served by that region. I have clarified this in the second paragraph of the Discussion, which interprets the correlation of BA45 activation with coherence.

It seems as though responses are scored as coherent if people say something similar to what the other participants have said on the matter (e.g., if people describe mornings, a good score will not be achieved if one has an unusual morning ritual). This would explain the decrease in "coherence" over the time-windows: people run out of common things to say after a few tens of seconds, like "getting out of bed", "taking a shower", "making coffee", etc. Staying on topic (i.e., saying what the others said) seems to be related to spending more resources (increased blood-flow) in the areas of interest. Or, people who have increased blood flow in these areas relative to when they say the humpty-dumpty verse, seem to be able to create more on-topic speech. So, is the current operationalization "coherence" appropriate? Well, it is one definition (amongst many that are possible), along the lines of a "staying-on-topic global coherence" measure and thus it would be interesting to see how the results would change if a different operationalization of coherence was used such as a "local coherence" metric (that arguably is more intuitively relevant) of how much the topics are "bouncing around" rather than what was used here. In general, the author needs to discuss how 'measure-specific' the current findings might be, namely to what extent they might be affected by subtle nuances in the natural language approach that has been adopted.

This is an important point, also raised by Reviewer 3. It is certainly true that coherence can be defined in a range of different ways. Researchers typically make a distinction between global and local coherence, as stated in the first paragraph of the Introduction. For the present study, I chose to focus mainly on global coherence, because this indexes the "off-topic" speech that increases in prevalence in old age. This is explained in the following new passage in the Introduction:

"Researchers have often made a distinction between local coherence, the degree to which adjoining utterances related meaningfully to one another, and global coherence, the degree to which each utterance relates to the topic under discussion (Glosser & Deser, 1992; Kintsch & Vandijk, 1978). Speech that is tangential or off-topic is therefore said to be low in global coherence, while speech that shifts abruptly between subjects is low in local coherence (although these measures are typically correlated). As most studies have reported that global coherence declines more severely than local coherence in later life

(Glosser & Deser, 1992; Kemper, Schmalzried, Hoffman, & Herman, 2010; Marini et al., 2005; Wright et al., 2014), this aspect of coherence was the focus of the present work.”

However, I agree it is important to consider whether the present findings generalise from the particular global coherence metric to other definitions of coherence. In response to this and other comments from the other reviewers, I have performed additional behavioural and neuroimaging analyses to investigate other properties of the speech produced in the scanner. These new analyses include a local coherence measure (computed using the same methods as the Hoffman et al. 2018 paper). I now report in the paper that:

1. Global and local coherence were positively correlated ($r=0.39$)
2. In a principal components analysis, global and local coherence both loaded strongly on the same latent factor, suggesting substantial covariance (Supplementary Table 3).
3. In a supplementary analysis that used the latent coherence factor rather than global coherence as a predictor of activation, results were very similar (Supplementary Figure 4). This indicates that the effects of coherence were not specific to the particular global coherence measure used in the main analysis; instead they generalised to a latent measure of coherence that is influenced by local, as well as global, coherence.

I hope these new analyses address the reviewer’s concerns about the measurement of coherence.

Results:

Another caveat to functional neuroimaging is that the results are based on the premise of cognitive subtraction such that the measured activation is a representation of the relative differences in brain activity between two or more brain states elicited by the task and therefore dependent on the intricacies of not only the task of interest but also the “baseline” conditions. Given the centrality of this issue, the onus is on the author to persuade readers that this ‘subtraction’ has been conducted appropriately. Specifically, regarding the temporal dynamics of coherence (i.e., coherence decreasing over time), could the 5s blocks of speech production be sensitive to pauses (e.g., containing no words, one word, etc.)? One could imagine that there are more pauses and fewer words per speech window at the end of the interval, creating a more labile estimate of coherence, and that this could be a factor in the clear trend of the impressive Supp.Fig.1. The author also mentions this factor on page 16, “Reduced activity over time was observed in primary auditory regions, perhaps due to a slowing of speech production later in the period.”. Even if this would probably not invalidate the overall results, it could be interesting to know the magnitude of this slowing, perhaps with an average of number of words produced per speech window.

This is a really good point. I have now analysed the number of words produced per block as function of the block's position in the speech period (this is presented in Supplementary Figure 1). In fact, there is no evidence for a reduction in words produced towards the end of the speech periods, as now reported in the Results:

"Supplementary Figure 1 also shows the mean number of words produced in each block. Participants tended to produce fewer words in the first 5s of each response but otherwise there were no systematic changes in speech rate over the 50s response period (the correlation between time in the response and number of words produced was 0.41 but this fell to 0.05 if the first block was excluded). In other words, there was no evidence that participants slowed their speech rate or ran out of things to say towards the end of the speech periods."

I have therefore removed the suggestion that negative time effect could be attributed to reduced speech rate.

There was also no relationship between number of words produced in a block and the coherence of the block ($r=-0.01$) as reported in the new section on other characteristics of speech.

p.17 - Figure 6 (and elsewhere) - the author refers to an activation pattern in 'highly coherent participants' versus 'less coherent participants'. Is it something specific to participants or to that specific speech sample? Put differently, is this effect that is documented consistent within the individual across testing times? Surely the latter must be demonstrated for the premise of the current manuscript to be valid as otherwise the phenomena being reported are more reflective of a more spurious issue associated with the stimuli used to generate the speech or some transient state-like phenomena...? Either way, this needs to be clarified given the premise of the study that speech coherence declines with age (and thus presumably this is what is intended to be measured).

Yes, this is an important point, addressed in the first paragraph of the Results:

"There was considerable variation in coherence across individuals: the least coherent individual had a mean coherence score of 41.5 and the most coherent 55.0. Importantly, the coherence measure showed high test-retest reliability across individuals. Fourteen of the participants had previously completed a similar speech elicitation task out of the scanner as part of an earlier study (Hoffman et al., 2018a). Their in-scanner and out-of-scanner coherence scores were highly correlated ($r = 0.88$), indicating that the observed variation represents stable individual differences in the ability to produce coherent speech."

Minor points:

P5, line 3: It would perhaps be more clear to state: "The main prediction of the current study was that...", or something similar, so the reader knows we have moved past the background information.

This section has been rewritten in response to R2's comments.

P5, line 14: "A recent study indicates...". Perhaps using past tense?

This section has been rewritten in response to R2's comments.

P8, and for results: It may be worth pointing out more clearly where the 20-word window approach was used, and where the 5s windows were used (approx. 10 words and below?). One can assume that the 20-word window was for the more overall measures for each response in total.

This has now been clarified in the Method. The 20-word window was used to assign a coherence value to each word in the response (the window for each word consisting of the word itself and the 19 words that immediately preceded it). Then, to obtain a coherence value to each 5s block, the coherence values for all the words in that block were averaged.

This is a slightly convoluted process but it has the advantage of holding the size of the window constant. The alternative – simply forming the windows out of the words produced in each 5s block – would have led to large variability in the size of the windows used (e.g., some with only 1 or 2 words and others with over 20). Pilot work had shown that this was undesirable, because coherence values based on very small windows tend to be unreliable.

P9, Figure 2, panel 3: How do we interpret the y-axis? Hemodynamic response?

Exactly right: this is a conventional fMRI analysis in which the dependent variable is the magnitude of the BOLD signal.

Additional comment on panel labels: Figure 4 is labeled with panels A-D, in Figure 2 the panels are labeled with 1-4 (with panel 3 legend referencing an A-C not present in the figure. This may perhaps be uniform between figures for clarity.

I agree that the points A-C in the legend are confusing and have removed these. I have retained that numbering of the panels as I think this best conveys the fact that they form a sequence of stages in the analysis.

P12, line 6: What was the method used to assess test-retest reliability, some variant of the ICC? Would be interesting to see the numbers, e.g., in a bracket after the sentence, with perhaps the second most reliable measure for reference.

Information about the test-retest reliability is given in the sentences following this statement, viz. "Fourteen participants had previously completed a similar speech elicitation

task out of the scanner as part of another study (Hoffman et al., 2018a). Their in-scanner and out-of-scanner coherence scores were highly correlated ($r = 0.88$), indicating that the observed variation represents stable individual differences in the ability to produce coherent speech.”

P21, third last line: Is “first investigation” perhaps a bit strong statement?

This statement has been reworded to remove the “first investigation” claim.

Reviewer #2

This is a very interesting study. The methods are clearly described. The statistics are appropriate. The results are clear and the discussion is well structured. I have only a few issues (mostly related to the introduction section and, partially, the discussion). Overall, I think that the paper needs a minor revision.

I am grateful for the reviewer’s encouraging and helpful comments.

Major issues:

1. Please provide a theoretical model for the statements at the beginning of the Introduction (e.g., the Framework Building Model by Gernsbacher, 1990 or Levelt’s model [1999]). It would be useful to appropriately introduce the hypotheses at the end of the Introduction and interpret the findings from this study with such psycholinguistic models in the Discussion.

Thank you for this suggestion. I have added discussion of these and other theoretical models towards the end of the Introduction. I hope that that this passage helps to embed the study in the broader psycholinguistic literature and to motivate the neural predictions. The new text reads as follows:

“The study predictions stemmed from the idea that to speak coherently, people must regulate their access to semantic knowledge so that they select the most relevant information to drive speech output. According to long-standing models of language comprehension, when people *comprehend* speech or text they generate a mental model of its content, often termed a situation model (Gernsbacher, 1991; Graesser, Singer, & Trabasso, 1994; Zwaan & Radvansky, 1998). This situation model is informed by the individual’s prior semantic knowledge about the topic under discussion. It is likely that a similar model-building process guides speech production (Garrod & Pickering, 2004; Kintsch & Vandijk, 1978; Levelt, Roelofs, & Meyer, 1999). In order to remain coherent, speakers therefore need to ensure that currently-relevant semantic knowledge contributes to the situation model guiding their production, while inhibiting irrelevant aspects of knowledge, which may lead to tangential or off-topic speech (Arbuckle & Gold, 1993; Marini & Andretta, 2016; Mozeiko, Le, Coelho, Krueger, & Grafman, 2011). The key prediction was

therefore that coherence in speech would be correlated with activation in brain regions that regulate access to semantic knowledge, chiefly the left inferior frontal gyrus (IFG).”

2. Before introducing the statement that “more coherent individuals perform better on tests of cognitive and executive control” it would be important to highlight that according to Miyake et al. (2000) and Mozeiko et al. (2011) executive functions are likely involved in different stages of message production (especially in the phases of message planning and framework laying).

I agree that this is an important point and have added a reference to Mozeiko et al. (2011) in the passage above.

3. Still in the Introduction, page 3: “Left IFG has been identified as the key node in a network of regions that control the retrieval and selection of semantic information”. In support of this claim, here the authors should also mention that an rTMS study by Marini and Urgesi (2012) suggests that the dorsal aspect of the left IFG is involved in the process of extraction of informative words during a discourse generation task. The inhibition of this area reduced the amount of informative words by increasing the production of off-topic utterances.

I agree and apologise for not citing this important study. It is now cited where the reviewer suggests.

Minor points:

1. Introduction, page 1: “coherence, the degree to which each utterance relates to the topic under discussion” here something is missing. Please check.

This has been corrected as follows: “the degree to which each utterance relates to the topic under discussion”.

2. Methods: Supplementary Table 1 - please provide the complete names (not just the acronyms) of the used tests in a legend. Also, please provide more information about the possibility that some individuals were under the age-related cut-off for normality. In case they were all over such cut-off just state it in the body of the text.

This information has been added to the legend to the table (all participants scored above the 10th percentile for their sex, age and education level on all tests).

Reviewer #3 (Remarks to the Author):

The manuscript presents a novel study investigating the effects of speech coherence on neural activation measured with fMRI.

The study topic is novel and the manuscript well written and would be of interest to the audience of Nature Communications and it is great that the

data is made available and methods presented very openly. However, I am not convinced that the measure of coherence (presented on its own as currently) is very informative. If I understand the ms correctly, the measure of coherence relies on similarity between an individual's utterances with the group (excluding the individual) composite scores. This seems like a very restricted definition of coherence.

I am grateful for the reviewer's encouraging and thoughtful comments on the ms. I respond to the specific comment about the coherence measure below.

My concerns include the following:

in at least one other behavioral paper by the author other measures of coherence are also presented (e.g., local coherence, based on the individual's own speech, I think) but not presented here; a range of other definitions of coherence could be arrived at, e.g., based on the whole corpus for each individual or based on similarity to clusters within the other subjects rather than the mean composite from the relatively small group or using methods other than LSA.

It is certainly true that coherence can be defined in a range of different ways. Researchers typically make a distinction between global and local coherence, as stated in the first paragraph of the Introduction. For the present study, I chose to focus mainly on global coherence, because this indexes the "off-topic" speech that increases in prevalence in old age. This is explained in the following new passage in the Introduction:

"Researchers have often made a distinction between local coherence, the degree to which adjoining utterances related meaningfully to one another, and global coherence, the degree to which each utterance relates to the topic under discussion (Glosser & Deser, 1992; Kintsch & Vandijk, 1978). Speech that is tangential or off-topic is therefore said to be low in global coherence, while speech that shifts abruptly between subjects is low in local coherence (although these measures are typically correlated). As most studies have reported that global coherence declines more severely than local coherence in later life (Glosser & Deser, 1992; Kemper, Schmalzried, Hoffman, & Herman, 2010; Marini et al., 2005; Wright et al., 2014), this aspect of coherence was the focus of the present work."

However, I agree it is important to consider whether the present findings generalise from the particular global coherence metric to other definitions of coherence. In response to this and other comments from the reviewers, I have performed additional behavioural and neuroimaging analyses to investigate other properties of the speech produced in the scanner. These new analyses include a local coherence measure (computed using the same methods as the Hoffman et al. 2018 paper). I now report in the paper that:

4. Global and local coherence were positively correlated ($r=0.39$)

5. In a principal components analysis, global and local coherence both loaded strongly on the same latent factor, suggesting substantial covariance (Supplementary Table 3).
6. In a supplementary analysis that used the latent coherence factor rather than global coherence as a predictor of activation, results were very similar (Supplementary Figure 4). This indicates that the effects of coherence were not specific to the particular global coherence measure used in the main analysis; instead they generalised to a latent measure of coherence that is influenced by local, as well as global, coherence.

I hope these new analyses address the reviewer's concerns about the measurement of coherence.

It is also not clear that semantic coherence is driving the effects seen: e.g., what about differences in the number of syllables/utterances made, or speech errors or speed of speech produced, or level of semantic complexity or size of vocabulary used;

This is a very important point, also raised by Reviewer 1. I have added a new section to the Results that considers the relationship between coherence and a range of other characteristics of speech – including the number of words produced per 5s block and properties related to vocabulary and semantic complexity (unfortunately speech errors were too rare to be analysed formally). The major findings of these new analyses were as follows:

1. Correlations between global coherence and the other characteristics of speech never exceeded 0.2 in magnitude. In particular, the correlation with number of words produced was -0.01 (see Supplementary Table 3). Thus, there appeared to be no major confounds between coherence and other aspects of speech.
2. In a principal components analysis of all the characteristics of speech, four latent factors emerged (Supplementary Table 4). These appeared to correspond to the following aspects of speech production:
 - a. Use of complex vocabulary (long, low frequency, late-acquired words)
 - b. Use of highly specific terms (words low in semantic diversity and high in concreteness)
 - c. Coherence (both global and local coherence strongly loaded this factor)
 - d. Verbosity (high number of words and high proportion of closed class words)
3. In a supplementary neuroimaging analysis, scores from all four latent factors were simultaneously entered as predictors of activation (along with time). This allowed me to test whether effects of the coherence factor would be found when controlling for other aspects of speech production. The results (shown in Supplementary Figure

4) closely resembled those of the main analysis: significant clusters were again found in left and right IFG and in RLFPFC.

In summary, the additional analyses indicate (a) that the global coherence measure used in the paper was not confounded with other properties of the speech produced and (b) similar effects of coherence on brain activation were present when other aspects of speech production were entered as covariates in the analysis.

equally, what about non-speech related issues such as effects of fatigue, background knowledge/life experiences etc.

Since this comment refers to differences between participants, it is relevant to the individual differences effects shown in Figure 6 and not the main coherence analyses (which looked at effects of coherence within-subjects). I agree that there are potential confounding factors here. It was not possible to control for all of these factors in the study and for this reason I de-emphasised these findings in the ms (i.e., I did not mention them in the Abstract and they were interpreted as a minor, preliminary finding in the Discussion). I believe the individual differences effects are of potential interest to readers and should remain in the paper; however, I agree caution is warranted and have added the following additional caveat to the Results where these findings are described:

“It is important to note that there are a number of other factors that may differ between more and less coherent speakers – for example, educational level, age, general knowledge and level of cognitive function – and these factors were not controlled for. Therefore, while this analysis identifies regions where activation varied between high and low coherence speakers, it is not possible to determine whether these effects are a direct consequence of coherence or are mediated by other factors.”

The manuscript would be greatly improved if other factors could be accounted for as much as possible. E.g., speech rate/semantic/syntactic errors/repetition as well presenting converging evidence from different definitions of coherence, including those derived from an individual's own language as much as possible.

I agree and I hope the additional analyses described above have addressed the reviewers' concerns.

REVIEWERS' COMMENTS:

Reviewer #1 (Remarks to the Author):

The author has produced a careful response to my comments. I appreciate many of the additions but there are still some concerns.

I remain concerned about the notion of modeling a rapid and dynamic phenomenon in this manner and most sincerely hope that follow-on papers will use far more sophisticated time-series techniques instead of using time as a covariate. Of course brain imaging pictures always look really convincing, until we realize they don't (really) have error bars, so probably should be used as a potential guide, not necessarily as a particularly strong argument in terms of the dynamics of language coherence.

The operationalization of coherence as done in this study remains a topic that can be discussed extensively in the future. The current idea of using distances in a constructed semantic space to measure coherence leads to the relatively simplistic notion of 'long distances = bad, lot of closeness = good' which is of course too simple, when we consider for example the prose in poetry that probably can take huge leaps but absolutely not be related to pathology. The other end is also interesting, namely it may not be all well and good if everything is very close in semantic space, indicating mostly common semantic relations. In a nut shell, the story is likely magitudes more complicated than the current research showcases.

Figure 7: I am happy that this figure was introduced, as it can be difficult to internally visualize the "windows" used for calculating the similarities between the utterance and the average representation of typical responses. The "sliding window" procedure is well illustrated here. However, this seems to be the exact same figure the author has previously published [in DOI: 10.7554/eLife.38907], only with the arrows indicating the local coherence calculations removed. In the case of the current manuscript, the figure would be greatly improved if it demonstrated the actual 20 word window size, not 10 as was the case here. This figure will also be at odds with an interpretation of figure 1.2, where it could seem like a "skipping window" approach where Block 1 (0-5s) contains one set of words given a single coherence score, even if there is well below 20 words, then the next Block (5-10s) contains another set of words, etc. From reading the updated method section, one could interpret it so that the first three blocks would get the same coherence score, as they all would have to refer to the first 20-word window. One suggestion could be to update the Figure 7 with a timeline on the x-axis below the sentence, with indications of blocks, like in the figure 1.4. This may not be crucial to the manuscript as a whole, but since this is a good opportunity for the author to introduce these methods to a wide audience, it seems appropriate to make details like this correct and ensure that figures 1 and 4 tell the same story.

Typo - in line 415 - the first word "who" should either be deleted or changed to "who were".

Reviewer #2 (Remarks to the Author):

The author has satisfactorily answered to all of my previous concerns.

Reviewer #3 (Remarks to the Author):

The author has addressed my concerns in the revised manuscript with compelling additional analyses and discussion. I am happy to reccomend publication.

Reviewer 1

I remain concerned about the notion of modeling a rapid and dynamic phenomenon in this manner and most sincerely hope that follow-on papers will use far more sophisticated time-series techniques instead of using time as a covariate. Of course brain imaging pictures always look really convincing, until we realize they don't (really) have error bars, so probably should be used as a potential guide, not necessarily as a particularly strong argument in terms of the dynamics of language coherence.

I appreciate the reviewer's helpful earlier comments on the issue of temporal resolution when modelling coherence. In the previous revision, I provided justification for the approach I have taken; this information is an important addition to the paper. I agree there is potential for more sophisticated techniques to be developed in future, and I believe the present work establishes an important starting point for such endeavours.

The operationalization of coherence as done in this study remains a topic that can be discussed extensively in the future. The current idea of using distances in a constructed semantic space to measure coherence leads to the relatively simplistic notion of 'long distances = bad, lot of closeness = good' which is of course too simple, when we consider for example the prose in poetry that probably can take huge leaps but absolutely not be related to pathology. The other end is also interesting, namely it may not be all well and good if everything is very close in semantic space, indicating mostly common semantic relations. In a nut shell, the story is likely magitudes more complicated than the current research showcases.

I certainly agree many factors contribute the quality of a communicative act and coherence is not always the most important one. In some circumstances, it may be desirable to sacrifice some degree of coherence in order to tell a more interesting or entertaining story, for example.

I have discussed these issues in more detail in my 2018 eLife paper. Space prevents me from discussing these issues at length in the present paper; however I have added the following passage to the Discussion to acknowledge this important point:

"Finally, it is important to note that while coherence is often a desirable quality in communication, this is not true in all contexts. For example, if one's aim is to entertain one's interlocutor, it may be advantageous to adopt a less focused and more digressive narrative style, which will make for a more interesting story [7]. Nevertheless, there are many situations in everyday life in which information must be communicated succinctly, and where maintaining focus on the topic at hand is paramount."

Figure 7: I am happy that this figure was introduced, as it can be difficult to internally visualize the "windows" used for calculating the similarities between the utterance and the average representation of typical responses. The "sliding window" procedure is well illustrated here. However, this seems to be the exact same figure the author has previously published [in DOI: 10.7554/eLife.38907], only with the arrows indicating the local coherence calculations removed. In the case of the current

manuscript, the figure would be greatly improved if it demonstrated the actual 20 word window size, not 10 as was the case here. This figure will also be at odds with an interpretation of figure 1.2, where it could seem like a "skipping window" approach where Block 1 (0-5s) contains one set of words given a single coherence score, even if there is well below 20 words, then the next Block (5-10s) contains another set of words, etc. From reading the updated method section, one could interpret it so that the first three blocks would get the same coherence score, as they all would have to refer to the first 20-word window. One suggestion could be to update the Figure 7 with a timeline on the x-axis below the sentence, with indications of blocks, like in the figure 1.4. This may not be crucial to the manuscript as a whole, but since this is a good opportunity for the author to introduce these methods to a wide audience, it seems appropriate to make details like this correct and ensure that figures 1 and 4 tell the same story.

I am glad the reviewer feels that Figure 7 has improved the paper. I accept their point that the window size in the figure is at odds with the actual window size (20) used in the calculations. I have modified Figure 7 so that it now reflects the actual situation better. Regarding the relationship between Figures 1 and 7, the reviewer is correct that in this examples blocks 1 and 2 would receive the same coherence value (although not block 3 as this extends beyond the first 20 words). Indeed, in Figure 1 blocks 1 and 2 do have the same value. I have therefore opted to leave Figure 1 as it is.

Typo - in line 415 - the first word "who" should either be deleted or changed to "who were".

This has been corrected.